# Cysteine-Rich Secretory Proteins (CRISPs) from Venomous Snakes: An Overview of the Functional Diversity in a Large and Underappreciated Superfamily

**DOI:** 10.3390/toxins12030175

**Published:** 2020-03-12

**Authors:** Takashi Tadokoro, Cassandra M. Modahl, Katsumi Maenaka, Narumi Aoki-Shioi

**Affiliations:** 1Faculty of Pharmaceutical Sciences, Hokkaido University, Faculty of Pharmaceutical Sciences, Hokkaido University, Kita-12, Nishi-6, Kita-ku, Sapporo 060-0812, Japan; tadokorot@pharm.hokudai.ac.jp (T.T.); maenaka@pharm.hokudai.ac.jp (K.M.); 2Department of Biological Sciences, National University of Singapore, Singapore 117543, Singapore; dbscmm@nus.edu.sg; 3Department of Chemistry, Faculty of Science, Fukuoka University, 19-1, 8-chomeNanakuma, Jonan-ku, Fukuoka 814-0180, Japan

**Keywords:** CAP superfamily, ion channel blockage, salivary component, co-factors

## Abstract

The CAP protein superfamily (Cysteine-rich secretory proteins (CRISPs), Antigen 5 (Ag5), and Pathogenesis-related 1 (PR-1) proteins) is widely distributed, but for toxinologists, snake venom CRISPs are the most familiar members. Although CRISPs are found in the majority of venoms, very few of these proteins have been functionally characterized, but those that have been exhibit diverse activities. Snake venom CRISPs (svCRISPs) inhibit ion channels and the growth of new blood vessels (angiogenesis). They also increase vascular permeability and promote inflammatory responses (leukocyte and neutrophil infiltration). Interestingly, CRISPs in lamprey buccal gland secretions also manifest some of these activities, suggesting an evolutionarily conserved function. As we strive to better understand the functions that CRISPs serve in venoms, it is worth considering the broad range of CRISP physiological activities throughout the animal kingdom. In this review, we summarize those activities, known crystal structures and sequence alignments, and we discuss predicted functional sites. CRISPs may not be lethal or major components of venoms, but given their almost ubiquitous occurrence in venoms and the accelerated evolution of svCRISP genes, these venom proteins are likely to have functions worth investigating.

## 1. Introduction

The CAP protein superfamily (Cysteine-rich secretory proteins (CRISPs), Antigen 5 (Ag5), and Pathogenesis-related 1 (PR-1) proteins), occasionally called the sperm coating protein (SCP) or Tpx-1/Ag5/PR-1/Sc7 (TAPS) family, occurs in a wide range of organisms. This superfamily is defined by a common structural feature, the CAP/PR-1 domain, with a unique α-β-α fold. The CAP/PR-1 domain comprises approximately 150–160 amino acids and includes four signature sequences, as defined in the PROSITE Database (http://www.expasy.ch/prosite/):

CAP1, [GDER][HR][FYWH][TVS][QA][LIVM][LIVMA]Wxx[STN];

CAP2, [LIVMFYH][LIVMFY]xC[NQRHS]Yx[PARH]x[GL]N[LIVMFYWDN];

CAP3 (HNxxR); and

CAP4 (G[EQ]N[ILV]).

Most proteins in this superfamily have only one CAP/PR-1 domain; however, a few species of parasitic helminths have proteins with more than one [1]. The CAP superfamily is extensive. The Pfam database (v32.0) (http://pfam.xfam.org/) contains 20,748 sequences (Pfam ID: PF00188) from 5356 species, ranging from bacteria to eukaryotes, and 39 structures, including a number of identical molecules with different IDs (Protein Data Bank, http://www.rcsb.org/; accessed on 2 January 2020). Sequence information for members of this superfamily continues to grow with the advancement of high-throughput technologies, such as next-generation cDNA sequencing.

Secreted PR-1 proteins were the first known members of the CAP superfamily, described in 1970 from *Nicotiana tabacum* plants infected with tobacco mosaic virus [2]. The abundance of PR-1 proteins increases in tobacco leaves infected with various pathogens [3]. These early results indicated that PR-1 proteins are involved in plant systemic responses to disease. Overexpression of the *PR-1* gene results in increased plant resistance to fungi [4], oomycetes [3,5], and bacteria [6], but not to viruses [7]. Subsequently, PR-1 proteins were found ubiquitously distributed among plants. *PR-1* genes are also associated with abiotic stress responses [8,9,10,11,12], though their expression may also be independent of stress responses [13]. The broad-ranging functions of PR-1 proteins require further investigation, especially after the discovery of PR-1 receptor-like kinases, which may be involved in initiation of signaling cascades [14]. The current hypothesis is that PR-1 proteins possess antimicrobial activity, amplifying defense signals via sterols or effector binding.

Ag5 proteins are abundant in insect venoms and saliva, including venoms of vespids and fire ants [15], and in the saliva of blood-feeding ticks [16], flies [17], and mosquitoes [18]. As one of the major allergens in insect venoms, immunoglobulins from human victims cross-react with Ag5s in venoms of yellow jackets, hornets, and paper wasps [15,19,20]. The function of Ag5 in saliva proteomes of hematophagous arthropods may be to regulate the host immune system and to inhibit coagulation during feeding [21,22]. For example, Ag5s from blood-feeding insects, *Dipetalogaster maxima* and *Triatoma infestans*, strongly inhibit collagen-induced platelet aggregation by interaction with Cu^2+^, providing redox potential for catalytic removal of O_2_, and decreasing inflammation [23].

CRISPs are highly expressed in rodent male reproductive tracts [24,25], with lower levels of expression in neutrophils, plasma, salivary gland, pancreas, ovary, thymus, and colon [26,27]. There has been a lack of consistent nomenclature regarding CRISPs. For example, CRISP-3 localized in seminal plasma is also known as specific granule protein 28 (SGP28), horse seminal plasma protein-3, and Aeg2 (NCBI Gene ID:10321). Three predominantly mammalian CRISPs (CRISP-1 to -3) have been referenced by different names in various studies, and a list of all published nomenclature has been assembled in a review by Adam et al. [28]. Mammalian CRISPs are associated with reproduction, cancer, and immune responses [28,29]. In addition to these activities, CRISPs have been identified as toxins in venom glands of snakes, lizards, spiders, scorpions, and cone snails [30,31,32,33,34]. Interestingly, a CRISP similar to those found in snake venom was also described as a main salivary component of the parasitic Japanese river lamprey (*Lethenteron japonicum*) [35].

CRISPs first appeared in reptile venoms approximately 170 million years ago in the clade Toxicofera [36,37]. Many CRISP orthologs have been found in lizard and snake venoms [38]. A review of venom proteomes confirmed the presence of CRISPs in viperid, elapid, and colubrid venoms, and their absence in atractaspidid venoms and those of some elapids, such as coral snakes [39]. The abundance of snake venom CRISPs (svCRISPs) in crude venom varies from 0.05% to 10%. The svCRISPs ablomin (*Gloydius blomhoffii*), triflin (*Protobothrops flavoviridis*), latisemin (*Laticauda semifasciata*), and tigrin (*Rhabdophis tigrinus*) were some of the first characterized, and were also classified as helothermine-like venom proteins (helveprins). Helothermine is a CRISP isolated from the venom of the Mexican beaded lizard (*Heloderma horridum*) [40]. The most common svCRISP activity has been non-enzymatic inhibition of various membrane channels, but many other activities have also been observed [31,41]. Sensitive “-omic” analyses, predominately transcriptomics and proteomics, have identified a large number of svCRISPs. However, in most cases these proteins have not been isolated or characterized experimentally, and their targets and biological roles remain unknown.

Target binding to alter cellular signaling cascades is a common function of CRISPs and other proteins of the CAP superfamily. In this review, we detail interactions between CAP superfamily proteins and their targets. It is important to view svCRISPs in the larger context of the entire CAP superfamily in order to identify their potential functions in venoms. We review svCRISPs that have been characterized during the past 10 years, examining their molecular surfaces and identifying regions and residues that contribute to their diverse biological activities.

## 2. Structural Features of Cysteine-Rich Secretory Proteins (CRISPs)

The Protein Data Bank (PDB) contains various CAP superfamily tertiary structures. High-resolution crystal structures of svCRISPs reveal a common secondary structure that includes 16 conserved cysteine residues (Figure 1). CRISPs have two main domains, a CAP/PR-1 domain at the N-terminus and a cysteine-rich (CRD)/ion channel regulatory (ICR) domain at the C-terminus, connected by a hinge region. For structure descriptions in this section, we have used residue numbering from triflin and natrin, well-characterized svCRISPs with published structures (Figure 1).

### 2.1. CAP/PR-1 Domain

Most CRISP structures contain an N-terminal α-β-α sandwich composed of five α-helixes and eight β-sheets, with five conserved cysteine residues among the 162 residues. The crystal structures of CRISP CAP/PR-1 domains show high similarity with CAP/PR-1 domains in P14a, a plant pathogenesis group-1 protein [42], and Ves v5, from yellow hornet venom [19]. A metal-ion-binding site in the CAP/PR-1 domain of CRISPs is also well conserved (His60, Glu75, Glu96, and His115; triflin numbering). Human CRISPs contain a glycosylation site (Asn-X(except Pro)-Ser/Thr) and a glycosylated form exists [43], but very few svCRISPs share this feature [41].

### 2.2. Hinge Region

About 20 amino acids (positions 163–182) form the hinge region between the CAP/PR-1 and CRD/ICR domains. The hinge region includes two disulfide bonds.

### 2.3. Cysteine-Rich Domain (CRD)/Ion Channel Regulatory (ICR) Domain

Three disulfide bonds and a few short α-helixes are well conserved in the CRD/ICR domain (positions 183–221). The CRD/ICR domain of svCRISPs may be important for recognizing ion channels. This has been suggested because the ion-binding motif in kaliotoxin (KTX) and margatoxin (an α-KTX) from buthid scorpion venoms [44], as well as in ShTx and BgK from sea anemone venoms [45], have the same tertiary structure. These peptide toxins, comprising approximately 40 amino acid residues, display high affinity for voltage-gated potassium channels and calcium-activated potassium channels, and possible interaction sites with these target channels have been proposed.

## 3. CRISP Co-Factors

CAP family members perform various physiological functions by binding to small compounds and proteins in the characteristic concavity of the CAP/PR-1 domain. Plant PR-1 proteins and the yeast CAP proteins, Pry1 and Pry2, bind sterols and lipids to inhibit pathogen proliferation. Sterols are essential for eukaryotes and bacteria, and removing them from membrane surfaces of pathogens inhibits their growth and can even kill them [46]. Lipid-related functions of the CAP superfamily have been summarized by Schneiter et al. in two reviews [47,48].

CRISPs bind divalent cations (Zn^2+^, Ca^2+^, and Cd^2+^), heparin, small peptides (substrates for Tex31, a cone snail CRISP [32]), and proteins (receptors). Five crystal structures of svCRISPs (natrin, triflin, pseudetoxin, pseudecin, and stecrisp) have revealed the presence of divalent ions in the CAP/PR-1 domain (Table 1). In crystal structures of the elapid CRISPs, pseudechetoxin and pseudecin, the CRD/ICR domains and the N-terminal domains form a groove that narrows upon Zn^2+^ binding, consistent with the finding that Zn^2+^ likely influences target molecule recognition [49]. Two Zn^2+^-binding sites in natrin are responsible for slight conformational differences with and without Zn^2+^, detected in 3D structure comparisons [50].

Binding of divalent cations alters CRISP activity. Zn^2+^ enhances the binding of natrin to heparin, resulting in increased expression of adhesion molecules on endothelial cells (ECs). It has been proposed that the heparin-binding site is located opposite the Zn^2+^-binding site in natrin [50]. Ca^2+^ (1 mM) increases cleavage activity of Tex31 (from *Conus textile*) 5-fold, but this increase was not observed with Zn^2+^ or Mg^2+^ [32]. It is interesting that members of the salivary antigen-5/CAP family from hematophagous insects are Cu^2+^-dependent antioxidant enzymes in competition assays, although these proteins also bind other divalent metals (Mn^2+^, Ni^2+^, Co^2+^, and Zn^2+^) depending on their presence in the buffers used [23]. Thus, there is clear evidence that divalent cations affect bioactivity not only of svCRISPs, but also other members of the CAP superfamily. However, whether conformational changes induced by cation-binding are correlated with their activity requires corroboration.

## 4. Proteins That Bind to CRISPs

Human CRISP-3 occurs in seminal plasma at high concentrations (14.8 g/mL) [72], but its function remains unknown. CRISP-3 is a promising bio-marker candidate for prostate cancer because the concentration of this protein increases >50-fold in pre-malignant prostate lesions and in primary tumors compared to normal prostatic epithelium [73]. To understand the physiological activity of CRISP-3, molecules with which it interacts have been identified, namely a prostate secretory protein of 94 amino acids (PSP94) (also known as a β-microseminoprotein; MSP) [74] and α-1B glycoprotein in human plasma (A1BG) [75]. Both bind to CRISP-3 with high affinity (*K*_D_ = 6.28 × 10^−11^ M with PSP94 and *K*_D_ = 2.8 × 10^−9^ M with A1BG). Identification of interaction surfaces between CRISP-3 and these binding proteins from mammals has been a focus of attention due to the medical relevance of CRISP-3 [76,77]. However, experimental evidence has been limited to NMR titration and mutagenesis analysis [78,79]. Recently, we determined the structure of a complex between PSP94 and CRISP family proteins that provided insight into CRISP-3 binding [80].

Small serum protein-2 (SSP-2) was identified in the serum of *Protobothrops flavoviridis* as an endogenous inhibitor against triflin (svCRISP) [81]. We built a binding model by superimposing SSP-2 onto PSP94, because PSP94 and SSP-2 are structurally similar and interact strongly with triflin across species [82]. The previously published PSP94–CRISP-3 model based on NMR titration showed that the N-terminal Greek key motif and the C-terminal β8 strand of PSP94 interact with the N-terminal CAP/PR-1 domain and hinge region of CRISP-3, respectively, in a parallel manner [78]. Our structure is upside-down compared to the other model, but the same surface of PSP94 interacts with the concave CAP/PR-1 domain of triflin (Figure 2A). In addition to the β5 and β8 strands, other key structural elements of PSP94 involved in complex formation are likely to be conserved. In PSP94, the β1 and β8 strands at the N- and C-termini are aligned in a linear manner and form an edged binding surface, whereas the β1 and β5 strands of SSP-2 form the binding surface. SSP-2 has a shorter C-terminal region compared with PSP94, so the N- and C-termini of SSP-2 are located on opposite sides. Consequently, this is in contrast to the N- and C- termini of PSP94, which are located on the same side. We hypothesize that formation of a parallel β-sheet between the SSP-2 β5 strand and the triflin β4 strand may allow the SSP-2 β1 strand to fit into the cavity between the CAP/PR-1 and CRD/ICR domains of triflin, thereby blocking the Zn^2+^ binding site and stabilizing the interaction. These findings indicate that our model provides significant structural insight into the human PSP94–CRISP-3 complex, which has been debated for many years.

The SSP-2–CRISP-3 complex model reveals that the N-terminal alanine of SSP-2 penetrates the metal-binding site of triflin, and that the CRD/ICR domain is shifted compared to the position of the CRD/ICR domain in triflin. This observation agrees with the conformational change of the CRD/ICR domain in the presence or absence of Zn^2+^, which has been documented for another svCRISP, pseudecine (Figure 2B). We have evidence that the binding of SSP-2 dramatically suppresses the channel inhibition activity of triflin (unpublished data). The structure of our complex also indicates that the concave region of the triflin CAP/PR-1 domain was fully occupied by the entire SSP-2 molecule, whereas direct interaction at the CRD/ICR domain was limited. Thus, SSP-2 may inhibit activity of several svCRISPs, because the concave region of this family shows great conservation. SSP-2 binding prevents Zn^2+^ binding to the concave region of the PR-1/CAP domain (Figure 2B). The structure-based alignment of venom CRISPs and human CRISP-3 show that the contact residues identified in our complex are relatively well conserved among CRISPs, suggesting the relevance of the binding ability of PSP94 to a wide range of CRISPs, including svCRISPs. As described above, divalent cations affect some CRISP activities (Section 3). Therefore, binding of the side face of the β-sheets of SSP-2 to both CAP/PR-1 and CRD/ICR domains of triflin might be important for suppression of triflin functions.

## 5. Isolation and Characterization of Snake Venom CRISPs

Snake venom CRISPs have proven difficult to express recombinantly and to fold properly in bacteria and yeast systems due to their eight disulfide bonds. Therefore, the first step in characterizing svCRISPs is usually to isolate from crude venom. Normal-phase and reversed-phase high-performance liquid chromatography (HPLC), have been used to purify venom CRISPs. The svCRISP natrin from *Naja*, triflin from *Protobothrops flavoviridis*, ablomin from *G. blomhoffii*, latisemin from *L. semifasciata*, tigrin from *R. tigrinus*, kaouthin-1 and kaouthin-2 from *Naja kaouthia* [83], and patagonin from *Philodryas patagoniensis* were purified by size exclusion chromatography, ion exchange chromatography, or heparin affinity chromatography. Pseudechetoxin from *Pseudechis australis*, TJ-CRVP from *Trimeresurus jerdonii*, and NA-CRVPs from *Naja atra* [84], as well as helothermine from the saliva of *Heloderma horridum* [40] and Tex31, from a homogenized extract of *Conus textile* [32], were purified by reversed-phase HPLC (RP-HPLC) with an acetonitrile buffer containing 0.1% (*v*/*v*) trifluoroacetic acid for the final purification step. It is possible that RP-HPLC purification in acidic buffers may affect CRISP activity and tertiary structure, although Tex31 which was purified by this method retained its proteolytic activity. In our own lab, we compared CD spectra of triflin purified by normal-phase and RP-HPLC and found no differences (unpublished). Therefore, we concluded that svCRISPs are very stable, even in strong acids (<pH 2.0).

In our work, two serum CRISPs were discovered in *Protobothrops flavoviridis* and *G. blomhoffii* venoms, which we designated as serotriflin and seroablomin, respectively [85]. We also reported complexes between serum CRISPs and other proteins in the blood of snakes. Formation of these complexes is pH-dependent, so they might not be found if acidic RP-HPLC buffers are used, but neutral or basic volatile buffer systems can probably be used.

The greatest challenge with CRISP isolation is that it is difficult to conclude if the biological activity is retained given that, for most, their activities are unknown. So far, no svCRISPs have proven lethal to mammals. A key feature of this family of proteins is that, although they bind various target molecules, all affect cellular signaling.

### 5.1. Ancestral CRISP Activity

Ito et al. and Xiao et al. characterized CRISPs from buccal glands of lampreys. These proteins included a CRISP from *L. japonicum* and buccal gland secretion protein-2 (CRBGP-2) from *Lampetra japonica* [35,51]. Lamprey CRISPs exhibited similar pharmacological effects under almost the same conditions and concentrations as svCRISPs, such as blocked depolarization-induced contraction of rat-tail arterial smooth muscle at 1 μM and suppression of angiogenesis related to EC apoptosis via microfilament disorganization [51,52,53,54], although their selectivity differed from that of svCRISPs [51]. Recombinant PR-1/CAP retained both cytotoxic activity against human umbilical vein endothelial cells (HUVECs) and anti-angiogenic activity. In 2011, a lamprey CRISP was demonstrated as a neutrophil inhibitory factor, and its inhibitory effect was caused by binding to α β 2 integrin receptors [55]. These observations suggested that some physiological activities of svCRISPs have been conserved from ancestral vertebrates.

### 5.2. Myotoxicity

Patagonin, from *Philodryas patagoniensis* venom, caused skeletal myotoxicity in murine gastrocnemius muscle, including muscle necrosis, edema, and inflammatory infiltration of polymorphonuclear leukocytes without smooth muscle contraction, as well as proteolytic activity, hemorrhage, and inhibition of platelet aggregation [70]. The authors hypothesized that the molecular mechanism by which patagonin induced muscle necrosis may be associated with binding to ion channels and speculated that this might be a general property of svCRISPs. They also suggested that tigrin (*R. tigrinus*) may cause skeletal myotoxicity because patagonin and tigrin are both from rear-fanged snakes and have high sequence similarity.

### 5.3. Ion Channel Inhibition

Several venom CRISPs from viperids and elapids target ion channels [31]. One of the best characterized svCRISPs is natrin, isolated from *N. atra*, the crystal structure and receptor targets of which are known. Natrin has an inhibitory effect on high-conductance calcium-activated potassium (BKca) Kv1.3 channels, as well as calcium release channel/ryanodine receptors (RyR) [67]. In 2010, Wang et al. demonstrated that <1 M natrin activated ECs to promote monocytic cell adhesion in a heparin sulfate- and Zn^2+^-dependent manner via increased expression of adhesion molecules (VCAM-1 and ICAM-1) and E-selectin as an inflammatory modulator [50]. They proposed that the mechanism involved binding of natrin to heparin in the presence of Zn^2+^. A cryo-EM study showed that the CRD/ICR domain of natrin is crucial for binding to ryanodine receptor 1 (RyR1, a Ca^2+^ release channel) [68]. However, sequence comparisons among svCRISPs suggest that the amino acid residues 42-44, 57-59, and 63-65 in the CAP/PR-1 domain may also be important for target channel recognition (Figure 1) [49,83]. These regions are putative interaction sites that target ion-channels, and are variable svCRISP residues. Indeed, the CRD/ICR domain of pseudechetoxin, a cyclic nucleotide-gated (CNG) channel blocker, did not inhibit CNG channels alone [49].

### 5.4. Anti-Protozoal Activity

Crovirin from *Crotalus viridis* has anti-protozoan activity against *Trypanosoma cruzi* and *Leishmania amazonensis* with low IC_50_ and LD_50_ values (1.10–2.38 g/mL), but was non-toxic to mice in an ex vivo assay measuring creatine kinase activity after an injection of 10 μg/mL of crovirin [71]. A considerably higher concentration of crovirin (20 g/mL) displayed limited toxicity to mammalian cells. The mechanism responsible for this activity was not investigated.

### 5.5. Anti-Angiogenic Activity

A CRISP from *Echis carinatus sochureki* venom, *EC*-CRISP, is a negative regulator of angiogenesis. The HPLC fraction did not interact with cancer cells, such as the glioma cell line, LN18, but showed pro-adhesive properties for normal ECs, such as HUVECs. At concentrations of 10–20 g/mL (<1 M), *EC*-CRISP interacted with ECs without affecting fibronectin, vitronectin, collagen type I, or laminin, and was transported into the cytoplasm. This toxin inhibited the MAPK Erk1/2 signaling pathway induced by vascular endothelial growth factor (VEGF), but had no effect on two other MAP kinases, p38 and SAPK/JNK [65]. However, the target receptor of ES-CRISP is still unknown.

### 5.6. Vascular Permeability Regulator

The effect of hellerin, from *Crotalus oreganus helleri* venom, on vascular permeability was demonstrated in vivo and in vitro. Trans-capillary leakage was observed 30 min after mice were subcutaneously injected with hellerin (70 nM), but leakage was approximately half that of the vascular permeability produced by vascular endothelial growth factor A (VEGF-A, 50 nM) [64]. Hellerin (2 M) reduced the viability of HUVECs by 50%, and hellerin-treated HUVEC cells had rounded cell shapes and detached from the substrate. In human dermal lymphatic endothelial cells (HDLECs) and human dermal blood endothelial cells (HDBECs), hellerin (675 nM) increased trans-epithelial permeability and decreased the level of F-actin.

### 5.7. Inflammation Regulator and Protease Activity

The *Bothrops jararaca* svCRISP, Bj-CRP, did not show proteolytic, hemorrhagic, coagulant, or potassium channel inhibition [63]. However, Bj-CRP increased leukocyte and neutrophil infiltration in mice 1 and 4 hr after i.p. injection, but this activity did not increase over 24 h following the injection. This inflammatory response might be related to the observed increase in IL-6 expression 1 h after injection, but no increase in TNF-, IL-10, or NO was observed. Bj-CRP cleaved C3 and C4 weakly and bound to components C3 and C4, which resulted in increased levels of C3a, C4a, and C5a. These results indicate that Bj-CRP modulates hemolytic activity associated with the complement pathways because >50 g of Bj-CRP reduced hemolytic activity.

## 6. Functional Sites Identified in CRISPs

CRISPs are highly conserved proteins in snake venoms; however, few svCRISPs have known biological activities (Table 1). Amino acid alignments between CRISPs with known structures and activities would be useful for understanding other CRISPs. CRISP putative functional sites have been proposed based on amino acid sequence conservation and variability. However, it is still necessary to characterize the biological activities of novel CRISPs because structure-function relationships for these toxins are yet to be completely understood.

### 6.1. Potential Functional Sites Responsible for Protease Activity

Cone snail Tex31, a PR-1 protein, showed proteolytic activity against synthesized peptides such as Ac-KLEKR-*p*NA. Tex31 is thought to cleave the propeptide region of conotoxin TxVIA as a native substrate [32]. Stecrisp, a svCRISP isolated from *Trimeresurus stejnegeri* venom, had no proteolytic activity under almost the same conditions. Based on tertiary structures, the authors proposed that Ser80 in Tex31, which is in a loop region, may be in close proximity to the highly conserved divalent cation site (His130 and Glu115 in Tex31), and this forms a catalytic triad responsible for cleavage [32]. Unlike Tex31, the residue at position 80 in stecrisp is proline (Pro80 in the reference [62]).

Patagonin and Lamprey CRISP lacked cleavage activity against fibrinogen [70]. Natrin lacked proteolytic activity against bovine serum albumin (BSA), neurotensin, a Tex31 substrate, or kenetensin [50]. Although Bj-CRP did not cleave azocasein, fibrinogen, or fibrin, it showed low catalytic activity against C3 and C4 [63]. This result might be similar to the general activation of component C3 by C3-convertase, a serine protease. C3-convertase cleaves component C3 to C3a and C3b, at a cleavage site between Arg726 and Ser727 (--LAR^726^S^727^NLD--) of component C3 [86]. This cleavage site is similar to the leucine at position P4 of the preferred substrate sequences of Tex31, counting from the C-terminal end. A sequence comparison between Bj-CRP and Tex31 could not be performed because the Bj-CRP sequence is not yet available. There are still many unknown sequences, such as the non-proteolytic patagonin. Thus, functional sites responsible for CRISP proteolytic activity are unclear because of the limited number of reports, and because their proteolytic activity against different substrates has not been comprehensively evaluated. Likewise, protease activity of mammalian, fungal, and plant CAP superfamily proteins is still lacking.

### 6.2. Potential Domains and Functional Sites Responsible for Ion Channel Inhibition

Seven CRISPs inhibit high-potassium-induced contraction of smooth muscle, with three of these (piscivorin, ophanin, and catrin) demonstrating a weaker effect at 1 µM [61]. Three CRISPs (tigrin, patagonin, and natrin) did not show inhibition when tested (Table 1). The current hypothesis is that suppression of muscle contraction is a result of the restriction of the Ca^2+^ incurrent by CRISP channel blockade; however, there is no direct evidence to support this. As previously discussed, the CRD/ICR domain is one of the CRISP regions potentially responsible for targeting ion-channels. This has been suggested because this domain is very similar to peptides that block ion channels, including peptide toxins KTX, α-KTX, and ShTx (43). CRD/ICR domain sequences between CRISPs that cause muscle contraction and those that inhibit contractions were compared (Figure 3A). Interestingly, there is a deletion of two residues in tigrin and an insertion of five residues (GAGGT) in lamprey CRISP in this region. These differences are also present only in the surface-exposed ICR motif of the CRD/ICR domain. (Figure 3B). However, there are no significant feature differences among active ion channel inhibitors and those that are inactive. There are likely multiple residues responsible for this activity in the CAP/PR-1 domain and/or hinge region.

Helothermine and natrin bind to ion channel targets with high affinity (<1 µM). These targets include K^+^ channels, ryanodine receptors [58,67,69], and voltage-dependent Ca^2+^ channels [57]. Not all CRISPs have been identified as having ion channel targets. Bj-CRISP had no effect on 13 voltage-gated potassium channels tested using two-electrode voltage-clamping on *Xenopus* oocytes. CNG channels were identified as pseudechetoxin and pseudecin targets. Interestingly, these toxins had different affinities, even though they differ by only seven residues. Two basic residues in pseudechetoxin (Lys184 and Arg185) seem to contribute to higher CNG binding affinity; the affinity of pseudechetoxin is 10-fold greater than that of pseudecin, with neutral residues (Asn184 and Tyr185) in this region (Figure 1) [60]. Matsunaga et al. discussed sequence variation in the concave region between the N- and C-terminal domains of svCRISPs. They suggested that variation in svCRISP activity toward different ion channels can be explained by charge distribution differences on the surfaces of svCRISPs [83]; however, there is no experimental evidence identifying the binding sites of these toxins.

## 7. svCRISP Evolution

CRISPs of toxicoferan reptiles have experienced positive selection, and more in snakes than in lizards [87]. In contrast, mammalian CRISPs appear constrained by negative selection. Episodes of rapid gene divergence are seen for svCRISPs in elapids and rear-fanged colubrids at all codon positions, in comparison to weaker gene divergence in viperid and boid snakes, and anguid, helodermatid, and iguanid lizards [88]. Evolution of toxicoferan CRISPs may be linked to snake predatory behavior. Elapid CRISPs manifest lower levels of positive selection (ω values), potentially due to the presence of highly toxic neurotoxins in these venoms for prey incapacitation. Viperids and rear-fanged colubrids have comparatively fewer toxic venom components, and have CRISPs manifesting higher levels of positive selection. At the protein-level, positive selection was identified at sites on the molecular surface, primarily in the CRD/ICR domain [87]. Future characterizations of CRISP activities with their corresponding functional sites may provide better insight into their role in prey capture and how this impacts their evolution.

Although 1-6 gene copies are observed for CRISPs in snake genomes [89,90], CRISP gene positions in snake genomes appear to be conserved. For genomes of both *Naja naja* and *Crotalus viridis,* svCRISP genes are located on chromosome 1 [90,91], potentially close to the centromere. This genome location may promote sequence conservation and may explain why expression of some toxins varies for different snake families, such as is observed for phospholipase A_2_s and three-finger toxins, but svCRISPs are present at consistent levels in venoms of most venomous snakes.

Expression levels of toxin and non-toxin homologs of svCRISPs have been evaluated for one opisthoglyph and for proteroglyphs and solenoglyphs [89,90,91,92]. At least two CRISP genes are expressed in snake venom glands, but differences in expression levels of each are apparent. Molecular evolution and functionality may not be equal for these two genes. Therefore, interpretations based on genome sequences alone require caution. Expression levels should also be considered to avoid misrepresenting the significance of venom CRISPs.

## 8. Conclusions and Future Directions

Two decades have passed since the discovery of venom CRISPs [58]. Many are now known and have been investigated in various ways, including protein and nucleotide sequencing, and characterization of molecular weights and isoelectric points. A few have been functionally characterized. However, new technologies generating large amounts of sequence data have completely overwhelmed our capacity to functionally characterize the many venom constituents being reported. This is especially difficult when it has yet to become apparent what biological roles certain toxins provide in venoms, as we can only document the activities for which we assay.

It is possible for toxins to have conserved amino acid sequences, functions, and mechanisms associated with their structures. This information can provide insights into their surface features and even into specific residues involved in binding. Conserved features may preserve selectivity and/or specificity. Structures and functions of CRISPs have been reviewed here, including those from lampreys. Ion channel-blocking activity and/or disruption of EC conditions may be an activity of svCRISPs that has been conserved evolutionarily. Further, it is important to consider co-factors and interacting proteins that regulate these activities in order to comprehensively understand biological effects of svCRISPs. Venom proteins are remarkably stable, with high selectivity and affinity for targets. Exploring toxins, even non-lethal ones like CRISPs, can improve our understanding of how proteins target specific channels, receptors, or substrates. This can be useful for the development of therapeutics, not only to treat snake envenoming, but also other maladies.

svCRISP characterization faces several challenges. The first is that these proteins occur at low levels in crude venoms. Moreover, eight disulfide bonds make them difficult to biosynthesize in large quantities for biological assays or structure determinations. An additional struggle lies in designing proper assays to characterize svCRISP activity, as a wide diversity of activities have been documented. We propose that the range of activities known from the CAP superfamily as a whole should dictate assays to be used in characterizing novel svCRISPs. Functional characterization of CRISPs lags far behind the number of genomic, transcriptomic, and proteomic CRISP sequences. An integrated approach to study snake venoms is required. Researchers involved in ‘-omics’ need to collaborate with labs that specialize in structure-function relationships to execute more comprehensive studies. In the future, we encourage toxinologists to characterize svCRISPs functionally.

## Figures and Tables

**Figure 1 toxins-12-00175-f001:**
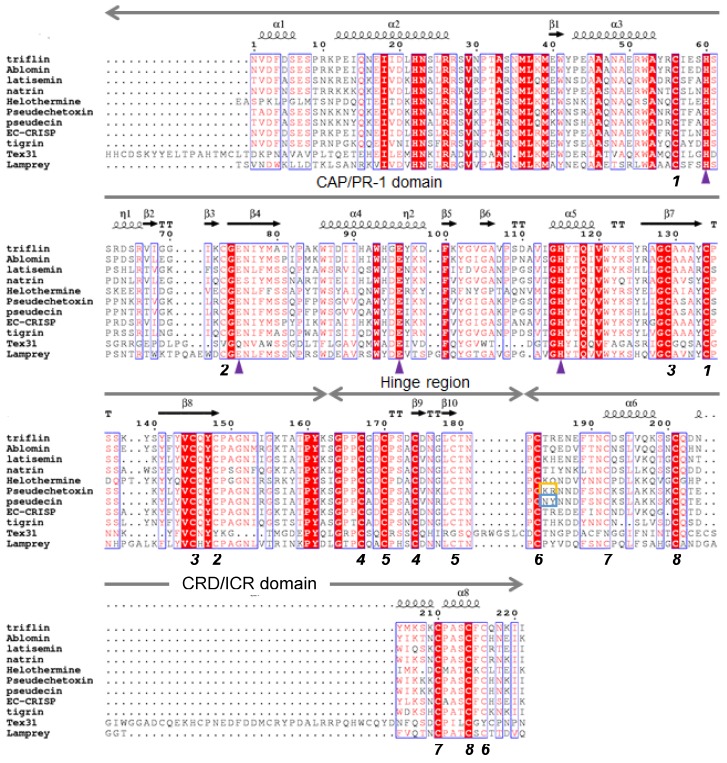
Amino-acid sequence alignments of Cysteine-rich secretory proteins (CRISPs). Highly conserved residues are highlighted in red, and other conserved residues are shown in a red font. Disulfide bridges are indicated below the alignment with black numbers. Identical numbers identify bonded residues. The secondary structure of triflin (PDB ID: 1WVR) is shown above the alignment. Purple triangles indicate conserved residues involved in binding of divalent cations (His60, Glu75, Glu96, and His115 for triflin). Basic residues in pseudetoxin are indicated with a yellow box, whereas the corresponding residues are neutral in pseudecin and are indicated with a blue box.

**Figure 2 toxins-12-00175-f002:**
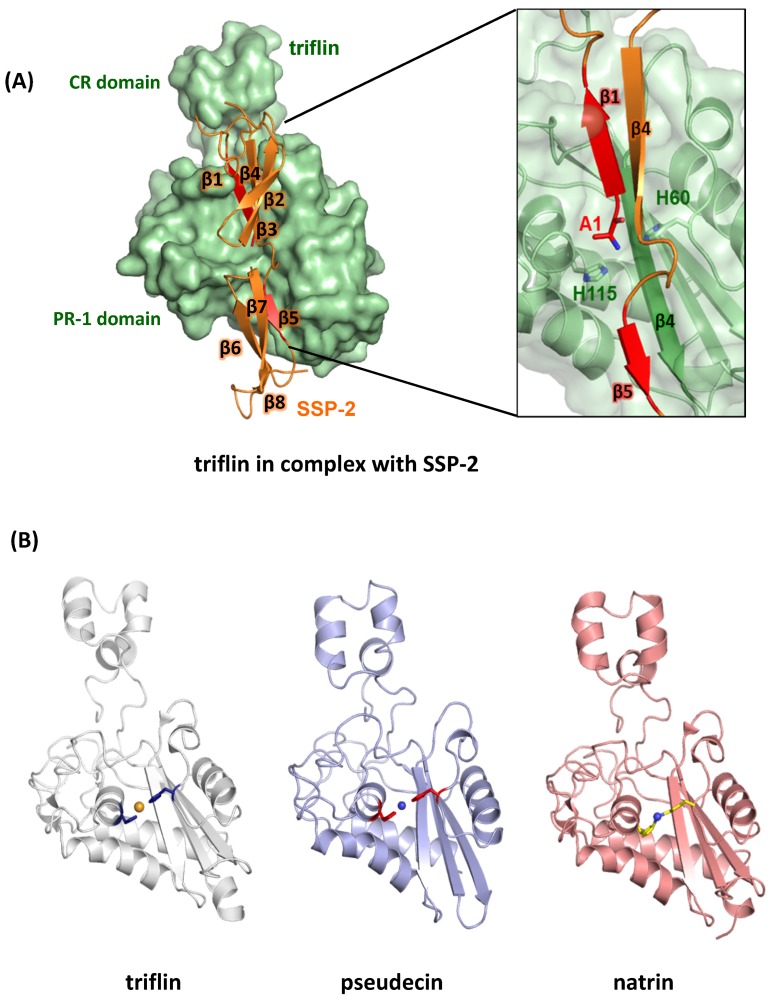
Inhibition of the divalent cation binding site by the serum inhibitor Small serum protein-2 (SSP-2). (**A**) Our complex structure of SSP-2-triflin (PDB ID: 6IMF) clearly indicates that the inhibitor occupies and blocks the conserved divalent cation binding site, which is functionally important. The inset is a focused view of the β1 and β5 strands of SSP-2. Ala1 of SSP-2 and His60 and His 115 of triflin are shown as stick models. (**B**) The same view of the apo-triflin structure (PDB ID: 1WVR, left), Pseudecin (PDB ID: 2FPF, middle) and natrin (PDB ID: 3MZ8, right) are shown. Divalent cations are bound at the conserved location via histidines, indicated with stick models. Structures were prepared using PyMOL (https://pymol.org/).

**Figure 3 toxins-12-00175-f003:**
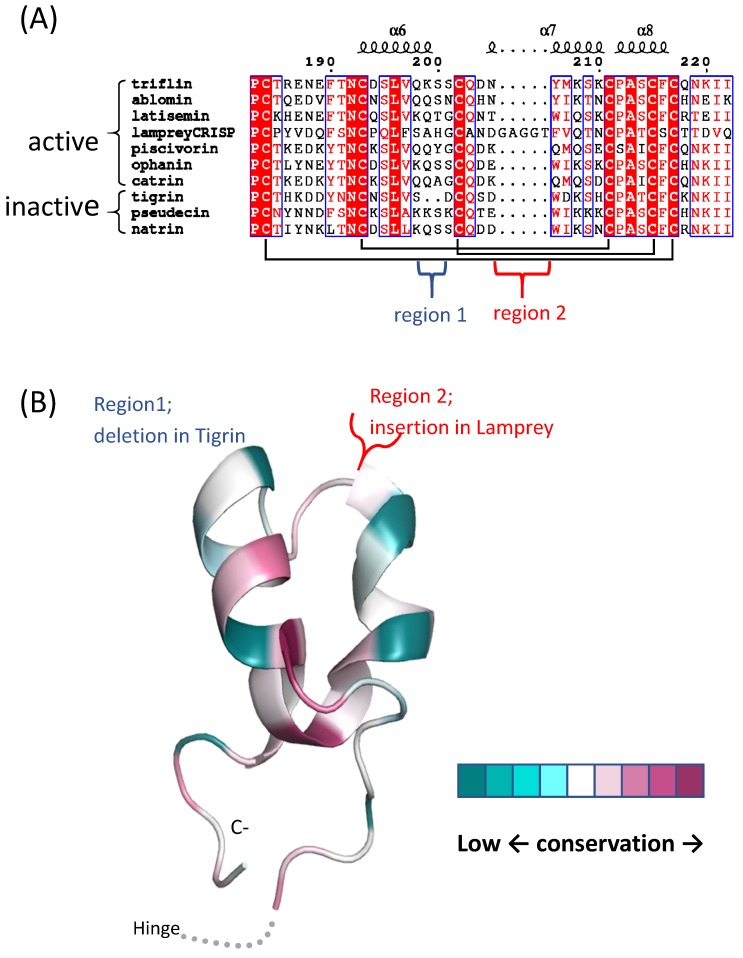
Comparisons between CRD/ICR domain regions of CRISPs that inhibit ion channels and those that do not. (**A**) Amino-acid sequence alignment of CRD/ICR regions of CRISP family proteins, showing highly conserved residues highlighted in red, and other conserved residues in a red font. Cysteine residues forming disulfide bridges are indicated by black brackets. The secondary structure of triflin (PDB ID: 1WVR) is shown above the alignment. (**B**) Structural conservation in the CRD/ICR region of snake venom CRISPS (svCRISPs0 that inhibit high-potassium-induced contraction of smooth muscle is shown on a triflin scaffold (PDB ID: 1WVR). Conservation scores were calculated with the Consurf server using default settings. Conservation scores are graded on a nine-point scale, from the most variable positions (turquoise) to the most conserved positions (maroon). The structure was prepared using PyMOL (https://pymol.org/).

**Table 1 toxins-12-00175-t001:** Targets and biological effects of cysteine-rich secretory proteins from buccal glands or venom.

Animals	Name	Species	Target (Interaction Molecules)	Biological Effect (or Related Investigation)	Accession Numbers	Ref.
Lamprey	Lamprey CRISP Buccal gland secretion protein-2 (BGSP-2) Cysteine-rich buccal gland protein (CRBGP)	*Lethenteron japonicum* *(Lampetra japonica)*	・Voltage-dependent Na^+^ channels ・Integrin β2 (CD11s/CD18)	Ca^2+^ channel blocker-like properties Anti-angiogenic activities Permeability Inhibition of adhesion, proliferation, migration, and invasion of cells (HUVEC; IC50 = 4.0 μM and Hela cell; IC50 = 6.7 μM) Non-fibrinogenolytic activity Activity of immunosuppressant (neutrophil inhibitory factor) Inhibition of Na^+^ channels in hippocampal neurons (12 μM) Inhibition K^+^ channels in hippocampal neurons (120 μM)	A4PIZ5	[35,51,36,37,38,39,40,41,42,43,44,45,46,47,48,49,50,51,52,53,54,55,56]
Cone snail	Tex31	*Conus textile*	N.D.	Proteinase	Q7YT83	[32]
Lizard	Helothermine	*Heloderma horridum* salivary secretion	・Ryanodine receptors ・Ca^2+^ channels ・K^+^ channels	Lethargy, partial paralysis of rear limbs and lowering of body temperature Blockage of receptors (Cerebellar Granule Cells) Inhibition of K^+^ channels (IC50 = 0.52 μM) Inhibition of Ca^2+^ channels (IC50 = 0.25 μM) Inhibition of skeletal ryanodine receptors (about 1.0 μM )	Q91055	[40,57,58,59]
Snake						
(Vipers)	Ablomin	*Gloydius blomhoffi.*	N.D.	Ca^2+^ channel blocker-like properties	Q8JI40	[60]
	Piscivorinc	*Agkistrodon piscivorus*	N.D.	Ca^2+^ channel blocker-like properties	AY181982	[61]
	Catrin	*Crotalus atrox*	N.D.	Ca^2+^ channel blocker-like properties	AY181983	[61]
	Triflin	*Protobothrops flavoviridis*	N.D.	Ca^2+^ channel blocker-like properties	Q8JI39 1WVR, 6IMF (with inhibitor)	[60]
	Stecrisp	*Trimeresurus stejnegeri* *(Viridovipera stejnegeri)*	N.D.	No proteolysis activity (unlikeTex31)	P60623 1RC9	[62]
	Bj-CRP	*Bothrops jararaca*	Component C3 and C4	Bind and cleaved to component C3 and C4 Lack of effect of K^+^ channel blockage activity (1.0 μM) (Kv1.1 to Kv1.6, Shaker-IR, Kv3.1, Kv7.1, Kv7.2, Kv7.4 and Kv10.1)	N.D. (partial sequence)	[63]
	Hellerin	*Crotalus oreganus helleri*	N.D.	To increase trans-epithelial permeability Cytotoxicity against HUVEC (cytotoxic concentration CC50 = 2.3 μM)	G9DCH4	[64]
	EC-CRISP	*Echis Carinatus Sochureki*	N.D.	Binding to HUVEC cell Anti-angiogenic Activities (10-20 μg/ml, <1 μM)	P0DMT4	[65]
	Crovirin	*Crotalus viridis viridis*	N.D.	Anti-protozoan activity against *Trypanosoma cruzi* and *Leishmania amazonensis*	N.D (partial sequence)	[66]
(Elapid)	Pseudechetoxin (PsTx)	*Pseudechis australis*	Cyclic nucleotide- gated ion channels	Inhibition of CNGA1 subunit (apparent *K*i = 70 nM) Inhibition of CNGA2 subunits (apparent *K*i = 15 nM)	Q8AVA4 2DDA	[49,67]
	Pseudecin	*Pseudechis porphyriacus*	Cyclic nucleotide-gated ion channels	Ca^2+^ channel blocker-like properties	Q8AVA3 2DDB, 2EPF(Zn2+)	[49,60]
	Latisemin	*Laticauda semifasciata*	N.D.	Ca^2+^ channel blocker-like properties	Q8JI38	[60]
	Ophanin	*Ophiophagus hannah*	N.D.	Ca^2+^ channel blocker-like properties	AY181984	[61]
	Natrin	*Naja Naja atra*	Bkca Kv1.3 Ryanodine receptors Heparin	High-conductance calcium-activated potassium (BKCa) channel (34.4 nM) Inhibition of Kv1.3 (10–200 nM) Inhibition of ryanodine receptors (1 μM, Kd = 1.5–2.3 nM) Inflammatory Modulator (<1.0 μM) Non-proteolytic activity (BSA, neurotensin, Tex31 substrate, kenetensin)	Q7T1K6 1XX5, 1XTA, 2GIZ 3MZ8 (Zn2+)	[68,69,70]
(Colubrid)	Tigrin	*Rhabdophis tigrinus*	N.D.	Non-Ca^2+^ channel blocker-like properties	Q8JGT9	[60]
	Patagonin	*Philodryas patagoniensis*	N.D.	Non-Ca^2+^ channel blocker-like properties Non-fibrinogenolytic activity Skeletal myotoxic activity (43 and 87 μM)	N.D. (only N-terminal sequence)	[71]

Tex31 is located in Pathogenesis-related 1 (PR-1) members. N.D.; not determined or did not investigate. Ca^2+^ channel blocker-like properties; inhibited depolarization-induced contraction of rat-tail arterial smooth muscle. All accession numbers are from UniProt (https://www.uniprot.org/) and Protein Data Bank (PDB, https://www.rcsb.org/pdb/home/sitemap.do).

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
