# Peer review of "Cysteine-Rich Secretory Proteins (CRISPs) from Venomous Snakes: An Overview of the Functional Diversity in a Large and Underappreciated Superfamily"

_toxins, 2020, doi:10.3390/toxins12030175_

Round 1

Reviewer 1 Report

The authors present a review on CRISPs. This paper is needed and informed me on many aspects of CRISPs I was unaware of. My perspective is snake transcriptome/genome focused and agree that CRISPs need much more investigation. Overall, the manuscript is well written and I only had a few minor changes to the text (in attached pdf). The one that needs to be fixed is having Crotalus viridis and crovirin in the colubrid section of Table 1. It is a viper and should be moved to the Viper section.

Suggestions for improvement that can be used or not, depending on what the authors want the scope of the paper to be.

1) For reviews, I like seeing a "Future road map" section or some equivalent title. The authors hint at some ideas for moving forward but it is not combined into a clear section at the end of the paper. A set of specific questions, hypotheses, goals, assumptions that could be tested for CRISPs would make this paper really nice. The authors do a good job setting up the problem (very little CRISP research) and summarizing what has been done, but the paper is missing more direct ways in which scientists can build on the current research to advance the knowledge quickly.

2) The current paper focuses on the proteomic side of CRISPs, hints at the transcriptomic research, and leaves out the genomic research. In the Crotalus viridis genome and the Indian Cobra Genome publications, they both find CRISP to be located on Chromosome 1. It may be close to the centromere and likely in a place within the genome that promotes sequence conservation. The genomic and transcriptomic data could be summarized and discussed more in the current review if the authors feel it is in the scope of the work.

Author Response

To Reviewer #1

We appreciate the constructive comments and expertise that the reviewer has provided. We have followed many of their suggestions in the revised manuscript. The changes are listed below, and marked in red throughout the manuscript.

 Response to reviewers comments;

(1) The one that needs to be fixed is having Crotalus viridis and crovirin in the colubrid section of Table 1. It is a viper and should be moved to the Viper section.

We have changed crovirin to be listed along with the other viper CRISPs.

(2)For reviews, I like seeing a "Future road map" section or some equivalent title. The authors hint at some ideas for moving forward but it is not combined into a clear section at the end of the paper. A set of specific questions, hypotheses, goals, assumptions that could be tested for CRISPs would make this paper really nice. The authors do a good job setting up the problem (very little CRISP research) and summarizing what has been done, but the paper is missing more direct ways in which scientists can build on the current research to advance the knowledge quickly.

We have added another paragraph summarizing future direction at the end of session 8.0 on page 15. We have also retitled this section “Conclusions and future directions”. The new paragraph reads:
“There are several challenges that currently exist in the toxinology field for characterizing svCRISPs. The first is that these proteins occur in lower abundances in crude venoms, moreover, their multiple disulfide nature makes them difficult to recombinantly express to obtain large quantities for biological assays or structure determinations. An additional struggle lies in designing proper assays to characterize svCRISP activity, as a wide diversity of activities have been documented. We propose that at least by considering the larger CAP superfamily as a whole, assays guided by activities within this family could help to characterize novel svCRISPs. Basic functional characterization of CRISPs is greatly lacking, with the number of genomic, transcriptomic, and proteomic CRISP sequences exceeding the number of CRISPs with determined structure and activities. An integrated approach to study snake venoms is required, researchers involved in ‘omics’ might be having to also collaborate with labs that specialize in structure-function relationships to provide greater depth to the descriptive research being done. In this review, we hope to encourage toxinologists to consider the functional aspects of svCRISPs, such as their mechanism of action, including the residues and structural motifs responsible for their effects.”

(3) The current paper focuses on the proteomic side of CRISPs, hints at the transcriptomic research, and leaves out the genomic research. In the Crotalus viridis genome and the Indian Cobra Genome publications, they both find CRISP to be located on Chromosome 1. It may be close to the centromere and likely in a place within the genome that promotes sequence conservation. The genomic and transcriptomic data could be summarized and discussed more in the current review if the authors feel it is in the scope of the work.

Following the reviewer’s suggestion, we have added a paragraph in section 7.0 “Snake venom CRISP evolution” that discusses svCRISP gene sequences. This paragraph reads:
“Although gene copy number variation is observed for CRISPs, with snake genomes reporting 1-6 CRISP genes [89, 90], CRISP gene positions in snake genomes appear to be conserved. For both genomes of an Elapidae and Viperidae snake (Naja naja and Crotalus viridis, respectively), svCRISP genes were located on chromosome 1 [90, 91], potentially close to the centromere. This genome location could be promoting sequencing conservation and could explain why the expression of some venom toxins are quite variable for different snake families, such as what is observed for phospholipase A2s and three-finger toxins in Elapidae and Viperidae snakes, but svCRISPs are present in consistent levels in the venom of the large majority of venomous snakes.”

That's all.

Sincerely yours,

Reviewer 2 Report

The review titled "Cysteine-rich secretory proteins (CRISPs) from venomous snakes: An overview of the functional diversity in a large and underappreciated superfamily" provides a nice overview of the CRISPs, filling a very important gap. The article is well-organized and very well-written.

I have two suggestions

  1. Although the article is a ‘general review’, It would have been better if some form of search strategy used in article selection was described.
  2. A figure that illustrates the functional diversity of the svCRSPs would have been beneficial for the reader.

In addition,

In the table 1, first column: use either classification hierarchy name (e.g. class/ order/ family) or the general names of the animal groups ( e.g. correct names as ‘vipers’, ‘elapid snakes’ and ‘colubrid snakes’)

Author Response

To Reviewer #2

We appreciate the constructive comments and expertise that the reviewer has provided. We have followed many of their suggestions in the revised manuscript. The changes are listed below, and marked in red throughout the manuscript. Please confirm attachment file, "toxins735071_revised_NS.pdf".

Response to reviewer's comments;

(1) Although the article is a ‘general review’, It would have been better if some form of search strategy used in article selection was described.

We have added a paragraph summarizing a future road map for research in this field. This new paragraph is at the end of the session 8.0 on page 15. We have tried to address future goals and retitled this section “Conclusions and future directions” . This paragraph reads:

“There are several challenges that currently exist in the toxinology field for characterizing svCRISPs. The first is that these proteins occur in lower abundances in crude venoms, moreover, their multiple disulfide nature makes them difficult to recombinantly express to obtain large quantities for biological assays or structure determinations. An additional struggle lies in designing proper assays to characterize svCRISP activity, as a wide diversity of activities have been documented. We propose that at least by considering the larger CAP superfamily as a whole, assays guided by activities within this family could help to characterize novel svCRISPs. Basic functional characterization of CRISPs is greatly lacking, with the number of genomic, transcriptomic, and proteomic CRISP sequences exceeding the number of CRISPs with determined structure and activities. An integrated approach to study snake venoms is required, researchers involved in ‘omics’ might be having to also collaborate with labs that specialize in structure-function relationships to provide greater depth to the descriptive research being done. In this review, we hope to encourage toxinologists to consider the functional aspects of svCRISPs, such as their mechanism of action, including the residues and structural motifs responsible for their effects.”

(2) A figure that illustrates the functional diversity of the svCRSPs would have been beneficial for the reader.

Following the reviewer’s suggestion, we have added a graphical figure that summarizes svCRISP activities. The graphical figure of our review will be upload on the author system of Toxins journal after final arrangement.

(3) In the table 1, first column: use either classification hierarchy name (e.g. class/ order/ family) or the general names of the animal groups (e.g. correct names as ‘vipers’, ‘elapid snakes’ and ‘colubrid snakes’)

We have changed the group names indicated in the first column of table 1. We have replaced the snake families with the general term “snake” to match what we have listed for other rows (lamprey, cone snail, lizard). Additionally, we used, Viper, Elapid and Colubrid as a supplemental information to explain the order in the snake section.

That's all.

Sincerely yours,

Reviewer 3 Report

The authors comprehensively summarize physiological activities of snake venom CRISPs and discuss the predicted functional sites on the basis of crystal structure and sequence alignments.

Line 64: "maximais" ---> "maxima".

Figure 1:  "CR/ICR domain" ---> "CRD/ICR domain".

Line 243-244: Are there any differences in amino acid sequence and physiological activity between serotriflin and triflin?

Author Response

TO Reviewer #3

We appreciate the constructive comments and expertise that the reviewer has provided. We have followed many of their suggestions in the revised manuscript. The changes are listed below, and marked in red throughout the manuscript. Please confrim attachmet file "toxins735071_revised_NS.pdf".

Response to reviewer's comments;

 (1)Line 64: "maximais" ---> "maxima".

We have corrected it to Dipetalogaster maxima.

(2)Figure 1:  "CR/ICR domain" ---> "CRD/ICR domain".

We have corrected it to CRD/ICR domain.

(3)Line 243-244: Are there any differences in amino acid sequence and physiological activity between serotriflin and triflin?

The percentage of identity between triflin and serotirflin is 65%, and the physiological activity of sertriflin is not yet known. However, we are currently investigating this and preparing a manuscript forced on the physiological activity of serum CRISPs and toxin CRISPs in venomous snakes.

That's all.

Sincerely yours,
